# DIFUSCO-LNS: DIFFUSION-GUIDED LARGE NEIGHBOURHOOD SEARCH FOR INTEGER LINEAR PROGRAMMING

## ABSTRACT

Integer Linear Programming (ILP) is a powerful and flexible framework for modeling and solving a variety of combinatorial optimization problems. This paper introduces a novel ILP solver, namely DIFUSCO-LNS, which combines the strengths of carefully engineered traditional solvers in symbolic reasoning and the generative power of a neural diffusion model in graph-based learning for the Large Neighborhood Search (LNS) approach. Our diffusion model treats the destroy policy in LNS as a generative problem in the discrete $\{0, 1\}$-vector space and is trained to imitate the high-quality Local Branching (LB) destroy heuristic through iterative denoising. Specifically, this addresses the unimodal limitation of other neural LNS solvers with its capability to capture the multimodal nature of optimal policies during variable selection. Our evaluations span four representative MIP problems: MIS, CA, SC, and MVC. Experimental results reveal that DIFUSCO-LNS substantially surpasses prior neural LNS solvers.

## 1 INTRODUCTION

Combinatorial Optimization (CO) problems (including NP-complete or NP-hard ones) present a set of fundamental challenges in computer science for decades (Papadimitriou & Steiglitz, 1998). Many of those problems can be formulated in a generic Integer Linear Programming (ILP) framework, including supply chain management, logistics optimization (Chopra & Meindl, 2001), workforce scheduling (Ernst et al., 2004), financial portfolios (Rubinstein, 2002; Lobo et al., 2007), compiler optimization (Trofin et al., 2021; Zheng et al., 2022), bioinformatic problems (Gusfield, 1997), and more. Classic ILP solvers typically conduct a tree-style search with the Branch-and-Bound (BnB) algorithm (Land & Doig, 2010), which finds the exact solution by gradually reducing and finally closing the gap between the primal (upper) and dual (lower) bounds of the searched solutions. Many state-of-the-art open-source and commercial ILP solvers are of this kind, including SCIP (Achterberg, 2009), CPLEX (Cplex, 2009), and Gurobi (Gurobi Optimization, 2021). However, when the problems are very large, completely closing the primal-dual gap can be intractable. Hence, solvers for large ILP problems have been shifted efforts towards *primal heuristics* (Berthold, 2006b), which are designed for finding the best possible solutions within a limited time window. That is, those are primal ILP solvers, which do not guarantee to find the optimal solutions. Our work in this paper belongs to the category of primal solvers.

Large Neighborhood Search (LNS) is a heuristic-driven strategy that can find high-quality solutions much faster than pure BnB for large ILP problems (Ahuja et al., 2002). The process starts from an initial feasible solution, which is typically obtained using BnB with a limited time budget. Then the system iteratively revises the current solution by selecting a subset of the variables as the *unassigned* (or *destroyed*) ones in the next cycle of optimization while keeping the remaining variables unchanged. Heuristics used in such a neighborhood selection are called the *destroy heuristics*. How to obtain good heuristics for effective neighborhood selection has been a central focus of LNS-based ILP solvers.

Hand-crafted destroy heuristics include randomized (Ahuja et al., 2002), Local Branching (LB) (Fischetti & Lodi, 2003b), and LB-RELAX (Huang et al., 2023b). The common limitation of those methods is their heavy dependencies on the availability of domain-expert knowledge, which is costly

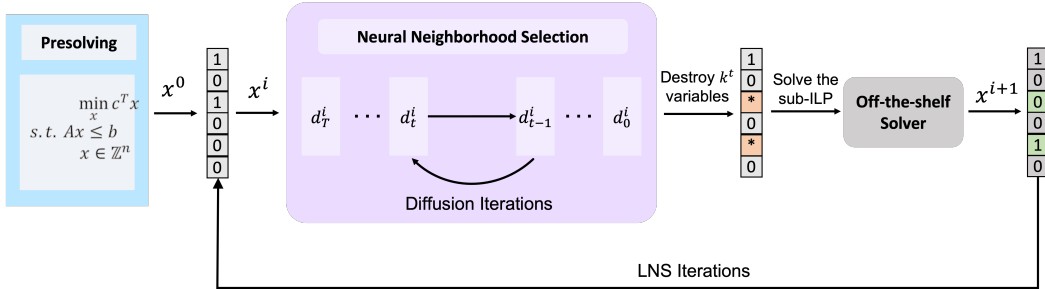

Figure 1: Overview of DIFUSCO-LNS's model framework.

to obtain, difficult to generalize across problems/domains, and unavoidably subjective sometimes. A better alternative, obviously, is to have a data-driven approach that can automatically learn the effective destroy heuristics from a training set of massive problem instances accompanied with high-quality (but not necessarily optimal) solutions. The recent development of neural network-based LNS methods has shown significant potential because they can learn from vast amounts of data. This ability allows them to identify complex patterns and relationships in combinatorial optimization problems, leading to the generation of more nuanced and effective destroy heuristics, surpassing the capabilities of traditional static methods.

Representative neural LNS methods include those based on Imitation Learning (IL-LNS) (Song et al., 2020b; Sonnerat et al., 2021), Reinforcement Learning (RL-LNS) (Nair et al., 2020a; Wu et al., 2021), and Contrastive Learning (CL-LNS) (Huang et al., 2023c). These methods essentially try to discover a good destroy policy for each instance ILP problem by predicting a discrete vector $\mathbf{d} \in \{0,1\}^{\|\mathcal{V}\|}$ with a conditional independence assumption among the variables. Here, $\mathcal{V}$ is the full set of candidate variables, $\sum d_i = k$ is the pre-defined size of the selected (or destroyed) neighborhood, and $d_i \in \{0,1\}$ indicates whether or not the $i^{th}$ variable is included in the neighborhood. The goal of prediction is to guide the search for optimal solutions being focused on the most promising sub-spaces of the feasible candidates. A fundamental limitation of the aforementioned neural LNS methods is in their implicit unimodal assumption in formulating the destroy policies. That is, it ignores the fact that multiple (near-)optimal destroy policies can co-exist with different subsets of destroyed variables (Li et al., 2018). In other words, those methods cannot properly handle the multimodal nature of the destroy heuristics in LNS. While this limitation is partially alleviated by reinforcement learning (Wu et al., 2021) and contrastive learning (Huang et al., 2023c), the current solutions still suffer severely from poor utilization of powerful neural networks due to the ineffective unimodal training.

In this paper, we address the above challenge/limitation from a new angle. We introduce DIFUSCO-LNS, as a pioneering effort to adapt the highly successful neural diffusion models from computer vision (Ho et al., 2020b; Song et al., 2020c; Song & Ermon, 2020) to the generation of effective destroy heuristics for LNS. Notably, neural diffusion models have also demonstrated proficiency in solving some other CO problems (Graikos et al., 2022; Sun & Yang, 2023; Huang et al., 2023a), but for using a probabilistic diffusion approach to solve generic Integer Linear Programming (ILP) problems, DIFUSCO-LNS is the first attempt, to our knowledge. It overcomes the limitation of previous neural LNS solvers with the power of handling the multimodal nature of high-quality destroy policies, in particular. Our diffusion model treats the destroy policy in LNS as a generative problem in the discrete $\{0,1\}$-vector space and is trained to imitate the high-quality Local Branching (LB) destroy heuristic through iterative denoising.

Our empirical evaluation demonstrates that DIFUSCO-LNS achieves a better or comparable performance against both neural baselines and traditional heuristics over four different CO benchmarks on multiple metrics. It also shows even stronger transfer performance (trained on small instances and tested on larger ones) than the state-of-the-art neural LNS method.

## 2 METHOD

### 2.1 PRELIMINARIES

Let us start with a brief outline of the related background of ILP and the techniques of LNS.

**Integer Linear Program** ILP is a type of discrete optimization problem whose variables are subject to integrality constraints. The general form of an ILP problem could be expressed as

$$
\begin{aligned}
&\min \mathbf{c}^\top \mathbf{x} \\
&\text{s.t. } \mathbf{A}\mathbf{x} \leq \mathbf{b}, \ \mathbf{x} \in \mathbb{Z}^n,
\end{aligned}
\tag{1}
$$

where $\mathbf{x} = (\mathbf{x}_1, \cdots, \mathbf{x}_n)^\top$ is the vector of decision variables, $\mathbf{c} \in \mathbb{R}^n$ is the vector of objective coefficients, $\mathbf{A} \in \mathbb{R}^{m \times n}$ and $\mathbf{b} \in \mathbb{R}^n$ represent the constraint coefficients . The size of an ILP problem is typically measured by its number of variables ($n$) and constraints ($m$).

**Neural Large Neighborhood Search** LNS is a process for iteratively improving the solution found by the system currently. It starts with an initial feasible solution $\mathbf{x}^0$, which is typically obtained by running a traditional symbolic solver with a limited time budget. In its $i$th iteration for $i = 0, 1, 2, \ldots, I$, the system heuristically chooses a subset of the decision variables in the current solution $\mathbf{x}^i$ as the destroyed (or unassigned) subset, and re-optimizes the next solution $\mathbf{x}^{i+1}$ over the destroyed variables while keeping the values of other variables unchanged. The re-optimization step is typically carried out by an off-the-shelf solver, and most research efforts in LNS have been focused on how to obtain good heuristics for the destroying part. Most neural LNS methods, including our proposed new approach in this paper, are focused on automated learning of such heuristics in a data-driven manner. As for the re-optimization part, we use the open-source SCIP solver (Achterberg, 2009) as the default symbolic solver. We denote the intermediate iteration state of LNS as $\mathbf{s}^i = (\mathbf{A}, \mathbf{b}, \mathbf{c}, \mathbf{x}^i)$.

**Local Branching** Local branching (LB) is proposed by Fischetti & Lodi (2003a) as a destroy policy heuristic (not a neural approach) for Large Neighborhood Search. It formulates the optimal neighborhood selection for LNS as another ILP problem, and searches for the next optimal solution $\mathbf{x}^{i+1}$ inside a Hamming ball with the radius of $k^i$ from the current incumbent solution $\mathbf{x}^i$. If all the decision variables are binary, solving LB is equivalent to solving the original ILP with additional constraint $\sum_{i=1}^n (1 - \mathbf{x}_j^i)\mathbf{x}_j + \sum_{j=1}^n \mathbf{x}^i(1 - \mathbf{x}_j) \leq k^i$. That is, LB itself is computationally expensive and could be practically intractable for finding optimal solutions in large-scale ILP. Therefore, it is essential to train a neural network to approximate the decisions made by LB with a much lower computational cost , thereby achieving real-world acceleration.

### 2.2 DIFUSCO-LNS

The goal of the neural destroy heuristic is to predict the destroy policy such that the new objective after the neighborhood search is maximized. In this paper, we adopt the supervised learning (i.e., imitation learning) scheme of neural LNS solvers. Following previous work (Sonnerat et al., 2021; Huang et al., 2023c), we use Local Branching (LB) as the expert heuristic to collect optimal (or high-quality) destroy policies.

We describe our approach in four parts: 1) probabilistic formulation of the LNS destroy policies, 2) diffusion-based modeling of destroy policies, 3) architecture of the policy neural network, and 4) automated generation of training data with time-constrained Local Branching.

**Problem Definition** We formulate the destroy policy of choosing the subset $\mathcal{V}^i = \{\mathbf{x}_{j_1}^i, \cdots, \mathbf{x}_{j_k}^i\}$ as a discrete vector $\mathbf{d}^i \in \{0, 1\}^{\|\mathcal{V}\|}$, where $\mathcal{V}$ is the full set of variables, $\sum d_j^i = k^i$ is the neighborhood size, and $d_i^j$ denotes the inclusion of the $j^{th}$ variable in the destroyed neighborhood at the $i^{th}$ LNS iteration. This allows us to formulate the destroy heuristic as a generative modeling problem, where we aim to maximize the likelihood of high-quality solutions. Let $\mathcal{D}_{\mathsf{hq}}^i$ be the set of high-quality solutions in the binary vector form, our loss function $L$ is defined as:

$$
L(\mathbf{s}^i, \boldsymbol{\theta}) = \mathbb{E}_{\mathbf{d}_{\mathsf{hq}} \in \mathcal{D}_{\mathsf{hq}}^i(\mathbf{s}^i)} \left[ -\log p_{\boldsymbol{\theta}}(\mathbf{d}_{\mathsf{hq}} | \mathbf{s}^i) \right]
\tag{2}
$$

For brevity, we omit the conditional notations of $\mathbf{s}^i$ and denote $\mathbf{d}_{\mathsf{hq}}$ as $\mathbf{d}_0$ as a convention for all formulas in the context of diffusion models.

**Generative Policy Modeling**  Following previous work on learning diffusion models that directly generate solutions for combinatorial optimization problems (Sun & Yang, 2023), we formulate the generation of the high-quality solution $\mathbf{d}_0$ as a discrete diffusion process (Austin et al., 2021; Hoogeboom et al., 2021).

The diffusion models first define a forward process $q$ that gradually corrupts[1] the data into noised latent variables $\mathbf{d}_1, \ldots, \mathbf{d}_T$: $q(\mathbf{d}_{1:T}|\mathbf{d}_0) = \prod_{t=1}^{T} q(\mathbf{d}_t|\mathbf{d}_{t-1})$. In discrete diffusion models with multinomial noises (Austin et al., 2021; Hoogeboom et al., 2021), the forward process is defined as: $q(\mathbf{d}_t|\mathbf{d}_{t-1}) = \mathrm{Cat}\left(\mathbf{d}_t; \mathbf{p} = \tilde{\mathbf{d}}_{t-1}\mathbf{Q}_t\right)$, where $\mathbf{Q}_t = \begin{bmatrix} (1 - \beta_{\mathsf{dm}}^t) & \beta_{\mathsf{dm}}^t \\ \beta_{\mathsf{dm}}^t & (1 - \beta_{\mathsf{dm}}^t) \end{bmatrix}$ is the transition probability matrix; $\tilde{\mathbf{d}} \in \{0,1\}^{N \times 2}$ is converted from the original vector $\mathbf{d} \in \{0,1\}^N$ with a one-hot vector per row; and $\tilde{\mathbf{d}}\mathbf{Q}$ computes a row-wise vector-matrix product. Here, $\beta_{\mathsf{dm}}^t$ denotes the corruption ratio. Also, we want $\prod_{t=1}^{T}(1 - \beta_{\mathsf{dm}}^t) \approx 0$ such that $\mathbf{d}_T \sim \mathrm{Uniform}(\cdot)$.

Next, a reverse (denoising) process is learned to gradually denoise the latent variables toward the data distribution, such that the distribution of $\mathbf{d}_0$ is formed as a joint distribution with latent variables:

$$p_{\boldsymbol{\theta}}(\mathbf{d}_{0:T}) = p_T(\mathbf{d}_T) \prod_{t=1}^{T} p_{\boldsymbol{\theta}}(\mathbf{d}_{t-1}|\mathbf{d}_t) \tag{3}$$

where $p_{\boldsymbol{\theta}}$ denotes a single reverse step parameterized by a neural network. According to Austin et al. (2021), the denoising neural network is trained to predict the clean data $p_{\boldsymbol{\theta}}(\widetilde{\mathbf{d}}_0|\mathbf{d}_t)$, and the reverse process is obtained by as an expectation over the posterior $q(\mathbf{d}_{t-1}|\mathbf{d}_t, \mathbf{d}_0)$:

$$p_{\boldsymbol{\theta}}(\mathbf{d}_{t-1}|\mathbf{d}_t) = \sum_{\widetilde{\mathbf{d}}} q(\mathbf{d}_{t-1}|\mathbf{d}_t, \widetilde{\mathbf{d}}_0) p_{\boldsymbol{\theta}}(\widetilde{\mathbf{d}}_0|\mathbf{d}_t) \tag{4}$$

By calculating the $t$-step marginal as: $q(\mathbf{d}_t|\mathbf{d}_0) = \mathrm{Cat}\left(\mathbf{d}_t; \mathbf{p} = \tilde{\mathbf{d}}_0\overline{\mathbf{Q}}_t\right)$, where $\overline{\mathbf{Q}}_t = \mathbf{Q}_1\mathbf{Q}_2 \ldots \mathbf{Q}_t$, the posterior we need at time $t-1$ can be obtained by Bayes' theorem:

$$q(\mathbf{d}_{t-1}|\mathbf{d}_t, \mathbf{d}_0) = \frac{q(\mathbf{d}_t|\mathbf{d}_{t-1}, \mathbf{d}_0)q(\mathbf{d}_{t-1}|\mathbf{d}_0)}{q(\mathbf{d}_t|\mathbf{d}_0)} = \mathrm{Cat}\left(\mathbf{d}_{t-1}; \mathbf{p} = \frac{\tilde{\mathbf{d}}_t\mathbf{Q}_t^\top \odot \tilde{\mathbf{d}}_0\overline{\mathbf{Q}}_{t-1}}{\tilde{\mathbf{d}}_0\overline{\mathbf{Q}}_t\tilde{\mathbf{d}}_t^\top}\right), \tag{5}$$

where $\odot$ denotes the element-wise multiplication, and $\mathbf{d}_0$ will be substituted by the predicted $\widetilde{\mathbf{d}}_0$ in the reverse process.

**Policy Network**  Recall that the general form of an ILP problem is represented by

$$\min \mathbf{c}^\top \mathbf{x} \quad \text{s.t.} \ \mathbf{A}\mathbf{x} \leq \mathbf{b}, \ \mathbf{x} \in \mathbb{Z}^n. \tag{6}$$

In DIFUSCO-LNS, the learned denoising neural network needs to encode the information of the last LNS iteration state $\mathbf{s}^i = (\mathbf{A}, \mathbf{b}, \mathbf{c}, \mathbf{x}^i)$ and the diffusion hidden states $\mathbf{d}_t^i$, and predict the high-quality destroy policy $\mathbf{d}_0$ as the output.

Expanding on recent advancements in learning for ILPs (Gasse et al., 2019; Sonnerat et al., 2021; Wu et al., 2021; Huang et al., 2023c), we adopt a bipartite graph representation to encode LNS state $\mathbf{s}^t$. This graph, composed of $n + m$ nodes, delineates the $n$ variables and $m$ constraints, with edges indicating non-zero coefficients in constraints. Node and edge features are inspired by Gasse et al. (2019). Moreover, a fixed-size window (size 3 in our experiments) of recent incumbent values enriches variable node features. Building upon previous work (Sonnerat et al., 2021; Huang et al., 2023c), we incorporate additional features from Khalil et al. (2017a) calculated at the BnB root node. The binary diffusion hidden state $\mathbf{d}_t^i$ is integrated as variable features with a positional encoding scheme (Vaswani et al., 2017; Sun & Yang, 2023).

---

[1] In the context of this work, we adopt specific terminologies for clarity: the destructive and reconstructive processes in LNS are termed as *destroy* and *repair* respectively, while the processes in diffusion models are designated as *corrupt* and *denoise*.

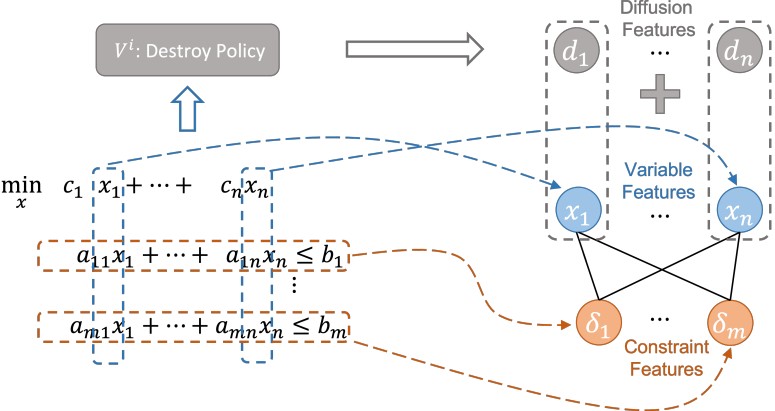

Figure 2: Architecture of the policy network

Following previous work (Huang et al., 2023c), the neural architecture of $p_{\boldsymbol{\theta}}$ is a graph attention network (GAT; Brody et al. (2021)). The detailed architecture design and hyper-parameters are described in the appendix.

**High-Quality Destroy Policy Collection**  When collecting the training data of high-quality destroy policies, given an intermediate iteration state $\mathbf{s}^i$, we use LB to find the (near)-optimal destroy policy $\mathcal{V}^i$ with a neighborhood size $k^i$ and a given time limit. If LB does not find $\mathcal{V}^i$ that leads to improved incumbent solution $\mathbf{x}^{i+1}$, we increase the neighborhood size in an adaptive manner (Huang et al., 2023b;c): At iteration $t$, when a better incumbent solution is found by LNS, the neighborhood size will be kept the same $k^{i+1} = k^i$, otherwise, it will be enlarged as $k^{i+1} = \min\{\gamma_{\mathsf{lns}} \cdot k^i, \beta_{\mathsf{lns}} \cdot n\}$, where $\gamma_{\mathsf{lns}} > 1$ is the size growing rate and $\beta_{\mathsf{lns}} \in (0, 1]$ controls the upper bound of the neighborhood size as a fraction of the total number of variables.

Upon solving the LB ILP, SCIP not only yields the (near)-optimal solution but also dumps the intermediate solutions encountered throughout the solving process. We target those intermediate solutions, denoted as $\mathbf{x}'$, that yield an enhancement in the objective value no less than a fraction $\alpha_{\mathsf{p}}$ of the maximum observed improvement, formalized as:

$$\mathbf{c}^{\mathsf{T}}(\mathbf{x}^i - \mathbf{x}') \geq \alpha_{\mathsf{hq}} \cdot \mathbf{c}^{\mathsf{T}}(\mathbf{x}^i - \mathbf{x}^{i+1}). \tag{7}$$

Such solutions are earmarked as high-quality expert policies. We impose an upper limit on the cardinality of the high-quality solution set $|\mathcal{S}_{\mathsf{hq}}^t|$ to $u_{\mathsf{hq}}$. In scenarios where the set exceeds this size, only the leading $u_{\mathsf{hq}}$ samples are retained, mitigating the potential solution degeneration. Following Huang et al. (2023c), the parameters were chosen as $\alpha_{\mathsf{hq}} = 0.5$ and $u_{\mathsf{hq}} = 10$.

## 3  EXPERIMENTS

### 3.1  EXPERIMENTAL SETUP

**Benchmark Datasets**  Following the previous evaluations in the literature (Sonnerat et al., 2021; Huang et al., 2023c), our evaluation used 4 benchmark datasets for a variety of synthetic CO problems, as listed in Table 1. Those problems include Maximum Vertex Covering (MVC), Maximum Independent Set (MIS), Combinatorial Auction (CA), and Set Covering (SC). We generate 1,000 small instances on each problem to collect the LB demonstrations for training. An additional 40 small instances and 40 large instances are used for evaluation, where the large instances contain twice as many variables as the small instances. To distinguish between evaluations on instances of varying sizes, we employ the '-S' and '-L' suffixes to indicate small and large instances respectively, as illustrated in Table 1.

We use the same procedure in (Huang et al., 2023c) to generate the training/testing instances and local branching demonstrations. We fix $k^0$ as 50, 500, 200, and 50 for MVC, MIS, CA and SC respectively. $\gamma$ is fixed as 1 on all datasets for LB. We use SCIP (version 8.0.1) to resolve ILP

Table 1: Statistics for the problem instances in each dataset. The number of variables and constraints are reported for instances in each dataset.

| Dataset | Small Instances | | | | Large Instances | | | |
|---|---|---|---|---|---|---|---|---|
| | MVC-S | MIS-S | CA-S | SC-S | MVC-L | MIS-L | CA-L | SC-L |
| # Variables | 1,000 | 6,000 | 4,000 | 4,000 | 2,000 | 12,000 | 8,000 | 8,000 1 |
| # Constraints | 65,100 | 23,977 | 2,675 | 5,000 | 135,100 | 48,027 | 5,353 | 5,000 |

formulated in LB and restrict the run-time limit for LB to 1 hour per iteration on each problem. An early stopping strategy is applied if there is no better incumbent solution found in three straight iterations.

**Baselines** We compare our methods to three non-neural baselines and two neural baselines featured by learning from LB, which include (1) BnB: the standard branch-and-bound algorithm used in SCIP (version 8.0.1), (2) Random-LNS: an LNS algorithm selecting the neighborhood with uniform sampling without replacement (3) LB-relax: an LNS algorithm which selects the neighborhood with LB-relax heuristic (Huang et al., 2023b) (4) IL-LNS: a neural LNS algorithm which learns LB heuristic through imitation learning (Sonnerat et al., 2021) and (5) CL-LNS: the state-of-the-art neural LNS baseline which learns LB heuristic via contrastive learning on both positive and negative samples (Huang et al., 2023c). We train all neural methods on the demonstration generated by LB on small instances and evaluate them on both small and large instances in the testing set. This amounts to in total 8 testing datasets, .

**Metrics** We use the common metrics[2] in previous evaluations for LNS methods and related baselines (Sonnerat et al., 2021; Nair et al., 2020a; Huang et al., 2023c), which include

1. *Primal bound* $\mathbf{c}^\top \mathbf{x}$: the objective value for the **feasible solution** $\mathbf{x}$.

2. *Primal gap* (Berthold (2006a)) $\gamma^p(\mathbf{x})$: the normalized difference between the primal bound and a pre-computed optimal (or best known) objective value $\mathbf{c}^\top \mathbf{x}$, defined as

$$\gamma^p(\mathbf{x}) = \begin{cases} 0, & \mathbf{c}^\top \mathbf{x}^* = \mathbf{c}^\top \mathbf{x} = 0 \\ 1, & \mathbf{c}^\top \mathbf{x}^* \cdot \mathbf{c}^\top \mathbf{x} < 0, \\ \frac{|\mathbf{c}^\top \mathbf{x}^* - \mathbf{c}^\top \mathbf{x}|}{\max\{|\mathbf{c}^\top \mathbf{x}^*|, |\mathbf{c}^\top \mathbf{x}|\}}, & \text{otherwise.} \end{cases}$$

3. *Primal integral* (Berthold, 2006a): the integral of the primal gap function $p(t)$ on the time range $[0, T]$. The primal gap function $p(t)$ is defined as the primal gap $\gamma^p(\mathbf{x}_t)$ for the best feasible solution $\mathbf{x}_t$ found until time $t$, and 1 if no feasible solution has been found yet.

### 3.2 MAIN RESULTS

We evaluate all methods on the synthetic datasets for MVC, MIS, CA, and SC problems. We basically follow the hyperparameter settings in (Huang et al., 2023c). For LB-relax, IL-LNS, CL-LNS, and DIFUSCO-LNS, we set $k^0$ as 100, 3000, 1000, and 100 for MVC, MIS, CA, and SC respectively. For IL-LNS, we compare the initial neighborhood size of 100 and 150 on SC and find $k^0 = 150$ works better in our experiment. For Random-LNS, $k^0$ is set as 200, 3000, 1500 and 200 for MVC, MIS, CA, and SC separately. We fix $\gamma = 1.02$ and $\beta = 0.5$ in the adaptive neighborhood size for LNS-based methods across all datasets. During the inference time, we first use SCIP to presolve a feasible solution $\mathbf{x}^0$ prior to the formal neighborhood search. The time budget for presolving is set as 10 seconds on all datasets except for SC-L, where we find a short-time presolving cannot find a decent initial solution so we extend its presolving time to 30 seconds. In our final results, we also filter out some instances that still suffer from bad initial solutions on SC-L and report the results on the remaining 30 instances. For each LNS iteration, we restrict the run-time limit of SCIP for the sub-ILP to 2 minutes.

---

[2]The optimal bound is calculated by the best method on each instance.

Table 2: Comparative result of all methods in the primal gap (PG) (in %) at 30-minute cutoff. We compare the average rank in the instance level for each method across both small and large datasets in the rightmost column. The best result is bolded on each dataset.

| | PG (%) ↓ | PG (%) ↓ | PG (%) ↓ | PG (%) ↓ | Avg. Rank |
|---|---|---|---|---|---|
| Dataset | MVC-S | MIS-S | CA-S | SC-S | |
| BnB | $2.64 \pm 0.36$ | $8.09 \pm 0.86$ | $3.02 \pm 0.74$ | $2.68 \pm 1.55$ | 5.36 |
| Random | $1.02 \pm 1.37$ | $0.20 \pm 0.18$ | $6.12 \pm 0.81$ | $3.13 \pm 1.50$ | 4.50 |
| LB-Relax | $1.21 \pm 1.4$ | $1.08 \pm 0.23$ | $6.61 \pm 0.98$ | $0.93 \pm 0.86$ | 4.31 |
| IL-LNS | $\mathbf{0.06 \pm 0.07}$ | $0.22 \pm 0.15$ | $0.34 \pm 0.36$ | $0.73 \pm 0.74$ | 2.64 |
| CL-LNS | $0.09 \pm 0.13$ | $0.24 \pm 0.15$ | $0.59 \pm 0.55$ | $0.86 \pm 0.99$ | 2.35 |
| DIFUSCO-LNS | $\mathbf{0.06 \pm 0.08}$ | $\mathbf{0.05 \pm 0.09}$ | $\mathbf{0.28 \pm 0.48}$ | $\mathbf{0.36 \pm 0.87}$ | **1.88** |
| Dataset | MVC-L | MIS-L | CA-L | SC-L | |
| BnB | $4.15 \pm 0.34$ | $8.04 \pm 0.34$ | $15.75 \pm 6.31$ | $3.11 \pm 1.78$ | 5.54 |
| Random | $0.48 \pm 0.22$ | $0.17 \pm 0.12$ | $6.44 \pm 0.78$ | $3.61 \pm 1.81$ | 4.31 |
| LB-Relax | $0.68 \pm 0.22$ | $5.43 \pm 0.26$ | $16.94 \pm 0.86$ | $\mathbf{0.46 \pm 0.82}$ | 4.02 |
| IL-LNS | $0.13 \pm 0.13$ | $0.13 \pm 0.11$ | $\mathbf{0.26 \pm 0.39}$ | $1.99 \pm 1.21$ | 2.59 |
| CL-LNS | $0.10 \pm 0.12$ | $0.27 \pm 0.16$ | $0.84 \pm 0.68$ | $1.11 \pm 1.22$ | 2.46 |
| DIFUSCO-LNS | $\mathbf{0.09 \pm 0.10}$ | $\mathbf{0.04 \pm 0.08}$ | $0.36 \pm 0.43$ | $0.77 \pm 0.78$ | **2.08** |

Table 3: Comparative result of all methods in the primal integral (PI) at 30-minute cutoff. We compare the average rank in the instance level for each method across both small and large datasets in the rightmost column. The best result is bolded on each dataset.

| | PI ↓ | PI ↓ | PI ↓ | PI ↓ | Avg. Rank |
|---|---|---|---|---|---|
| Dataset | MVC-S | MIS-S | CA-S | SC-S | |
| BnB | $58.69 \pm 5.64$ | $151.27 \pm 8.24$ | $142.46 \pm 31.86$ | $89.09 \pm 25.8$ | 5.39 |
| Random | $31.07 \pm 23.7$ | $19.01 \pm 3.01$ | $131.23 \pm 13.39$ | $86.96 \pm 25.5$ | 4.64 |
| Relax | $37.61 \pm 23.26$ | $46.42 \pm 4.57$ | $164.16 \pm 17.6$ | $43.18 \pm 15.26$ | 4.23 |
| IL-LNS | $\mathbf{12.40 \pm 1.50}$ | $17.80 \pm 2.55$ | $\mathbf{39.03 \pm 9.48}$ | $37.21 \pm 12.99$ | 2.49 |
| CL-LNS | $13.02 \pm 2.74$ | $18.53 \pm 2.61$ | $45.60 \pm 10.69$ | $40.93 \pm 15.91$ | 2.28 |
| DIFUSCO-LNS | $12.45 \pm 1.74$ | $\mathbf{15.83 \pm 1.86}$ | $42.81 \pm 8.46$ | $\mathbf{30.76 \pm 14.51}$ | **1.99** |
| Dataset | MVC-L | MIS-L | CA-L | SC-L | |
| BnB | $76.05 \pm 6.05$ | $151.05 \pm 6.1$ | $331.37 \pm 34.81$ | $125.15 \pm 23.89$ | 5.44 |
| Random | $27.45 \pm 3.68$ | $\mathbf{29.99 \pm 2.65}$ | $147.39 \pm 12.26$ | $153.33 \pm 210.22$ | 4.19 |
| Relax | $53.62 \pm 3.72$ | $131.6 \pm 4.44$ | $330.67 \pm 16.28$ | $64.18 \pm 17.60$ | 4.10 |
| IL-LNS | $16.76 \pm 2.56$ | $35.49 \pm 3.26$ | $31.09 \pm 7.19$ | $120.07 \pm 21.69$ | 2.36 |
| CL-LNS | $\mathbf{16.65 \pm 2.30}$ | $39.42 \pm 4.07$ | $42.64 \pm 10.85$ | $66.54 \pm 23.14$ | **2.29** |
| DIFUSCO-LNS | $16.96 \pm 2.11$ | $32.06 \pm 2.94$ | $\mathbf{25.64 \pm 6.98}$ | $57.47 \pm 15.98$ | 2.62 |

We compare the primal gap (PG) and primal integral (PI) at the 30-minute cutoff of all methods in Table 2 and Table 3, and visualize the change of the primal gap on each dataset in Figure 3. Please refer to the Appendix for additional results in the primal bound. We find the reproduced results for some baseline methods contradictory to the conclusion in (Huang et al., 2023c) due to the difference in computational resources, but we ensure a fair comparison among all methods under the same computational environment.

It can be seen that DIFUSCO-LNS achieves a better or comparable performance against all previous baselines in both the primal gap and primal integral. We compute the average rank for each method across either the small or large datasets, and DIFUSCO-LNS always owns the lowest average rank in either metric. In our experiment, we notice that a higher AUC-ROC in predicting LB's neighborhood selection does not necessarily translate into an improved primal gap or primal integral. A neighborhood selection closer to LB's choice typically leads to a larger improvement in the primal bound in a single step, nonetheless, the induced sub-ILP from this neighborhood selection could also take a longer time to solve. In comparison, the neighborhood selection leading to a small primal

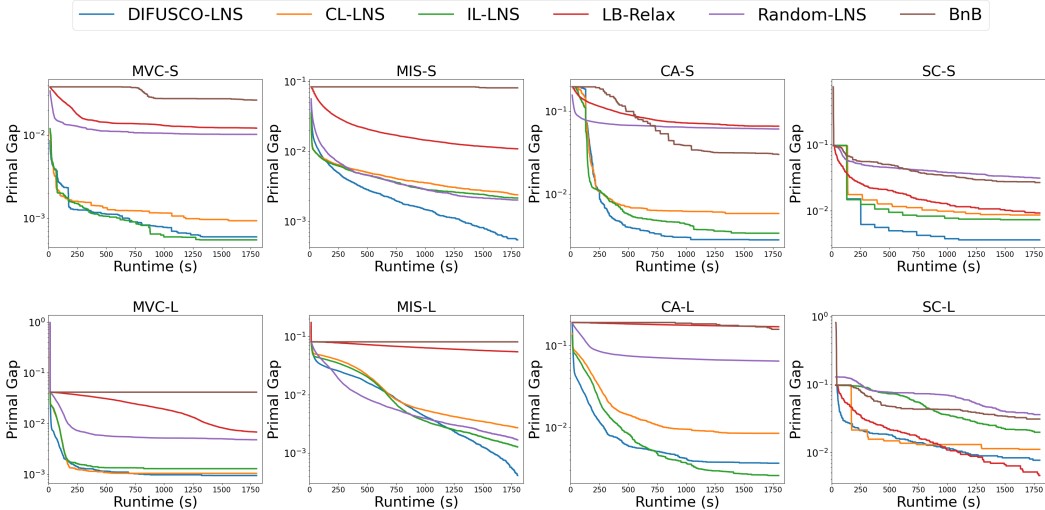

Figure 3: The plot of the primal gap (the lower is better) as a function of runtime on all datasets. For a more straightforward comparison between LNS-based methods and BnB, we clip the initial presolving stage (30 seconds for SC-L and 10 seconds for others) for all methods in the plot.

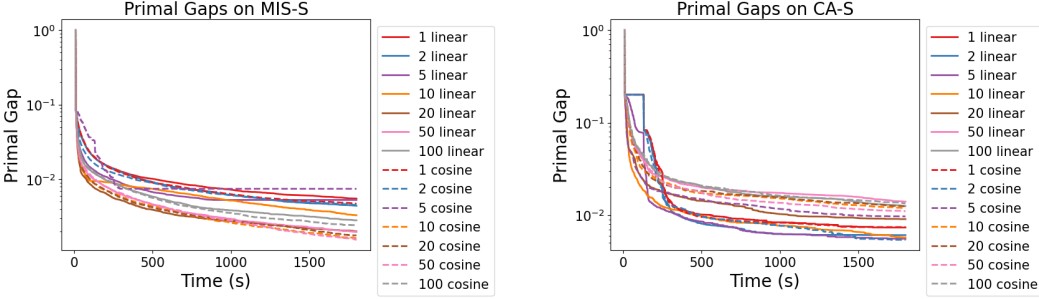

Figure 4: Primal Gap for DIFUSCO-LNS with a different number of inference diffusion steps and inference schedulers on MIS-S.

Figure 5: Primal Gap for DIFUSCO-LNS with a different number of inference diffusion steps and inference schedulers on CA-S.

bound improvement may instead create a simple sub-ILP solvable in a short time. Solving multiple such simple sub-ILPs can lead to a larger primal bound improvement in total and also a smaller primal integral than solving a single hard sub-ILP within the same solving time. A typical example is MIS-L where Random-LNS owns the best primal integral against all other neural methods, although DIFUSCO-LNS achieves the better primal gap at the end of the solving. Such observation is also consistent with the results from previous works like (Huang et al., 2023c). This explains why DIFUSCO-LNS sometimes shows an inferior performance to some weak baselines on certain datasets. But in most cases, DIFUSCO-LNS still makes a better prediction of the neighborhood selection from LB and translates it into the lower primal gap and primal integral.

## 3.3 ABLATION STUDY

Since DIFUSCO-LNS involves more parameters than previous neural baselines, we analyze the effect of these hyperparameters on DIFUSCO-LNS's performance in our ablation study. We choose its number of inference diffusion steps from the set $\{1, 2, 5, 10, 20, 50, 100\}$ and compare the linear and cosine inference schedulers on MIS-S and CA-S datasets. The primal gaps are visualized in Figure 3.3 and 3.3.

On both datasets, the optimal performance is achieved at the number of steps less than 100. Considering that 100 steps are still affordable, DIFUSCO-LNS actually does not suffer from the addi-

tional inference time from the sampling. In fact, we have observed that 1-step inference has already achieved a descent result on MVC, CA, and SC datasets. Besides, the performance differences caused by different inference schedulers or different numbers of inference steps are actually well-bounded, all inference hyperparameters can lead to a final primal gap around $10^{-2}$. This justifies that DIFUSCO-LNS is easy to tune and insensitive to the hyperparameter choices.

## 4 RELATED WORK

### 4.1 LEARNING PRIMAL HEURISTIC FOR MILP

The primal heuristics in combinatorial optimization aim to efficiently find high-quality feasible solutions. Diving and LNS are two main classes of primal heuristics and traditional solvers typically adopt a mixture of different variants of diving and LNS. Existing neural methods for primal heuristics mainly focus on the heuristics selection (Khalil et al., 2017b; Hendel et al., 2019; Chmiela et al., 2021), neural diving (Nair et al., 2020b; Yoon, 2022; Han et al., 2023a; Paulus & Krause, 2023) and neural LNS (Song et al., 2020a; Addanki et al., 2020; Sonnerat et al., 2021; Wu et al., 2021; Huang et al., 2023c).

LNS iteratively refines the solution by selecting a subset of variables, the neighborhood, to optimize at each time. Recent neural LNS methods mainly focus on the learning of neighborhood selection and leave the optimization for an off-the-shelf solver. (Song et al., 2020a) learn to partition the variables into subsets which then sequentially serve as the neighborhood to search in LNS. Later, (Wu et al., 2021) and (Addanki et al., 2020) propose more general RL frameworks directly predicting the variables to optimize at each iteration. Although Song et al. (2020a) also experiment with the imitation learning method, the training instances are obtained from random sampling which suffers from poor qualities. (Sonnerat et al., 2021) thus propose to utilize a strong expert heuristic *local branching* to generate high-quality demonstrations. Recently, CL-LNS (Huang et al., 2023c) adopted contrastive learning to learn from both positive and negative samples collected by local branching. In this work, we also aim to learn neighborhood selection from the local branching heuristics but instead rely on more powerful diffusion models.

### 4.2 DISCRETE DIFFUSION MODELS

Diffusion models (Sohl-Dickstein et al., 2015; Song & Ermon, 2019; Ho et al., 2020a; Song & Ermon, 2020; Nichol & Dhariwal, 2021; Karras et al., 2022) are widely used as the generative model for continuous data, which progressively adds the Gaussian noise to the real samples and learns the conditional denoising step in the reverse process.

Discrete diffusion models follow the same diffusion process but work on the discrete domain, such as text (Johnson et al., 2021; He et al., 2023), sound (Yang et al., 2023), protein (Luo et al., 2022) or molecule (Vignac et al., 2023). There are typically two ways to realize the discrete diffusion models. One type of method keeps the discrete structure by adding binomial (Sohl-Dickstein et al., 2015) or multinomial/categorical noise (Austin et al., 2021; Hoogeboom et al., 2021) directly to the discrete input. The other approach instead transforms the discrete data to the continuous space (Gong et al., 2023; Li et al., 2022; Dieleman et al., 2022; Chen et al., 2022; Han et al., 2023b) and then applies the standard diffusion models. Recently, Sun & Yang (2023) applied a graph diffusion model, DIFUSXO, on NP-hard problems and achieved remarkable improvement. In this work, we extend DIFUSCO to LNS which allows a more general application on CO problems.

## 5 CONCLUSION

In this paper, we propose DIFUSCO-LNS, a novel ILP solver that synergistically leverages the symbolic solving capabilities of carefully engineered traditional solvers with the generative power of diffusion models within the Large Neighborhood Search (LNS) framework. We evaluated our model on four representative MIP problems and found it is competitive, or outperforms the strong IL-LNS, CL-LNS, and LB-relax baselines. In the future, we are interested in accelerating the inference speed of diffusion models with more advanced diffusion solvers (Campbell et al., 2022; Sun et al.,

2022; Huang et al., 2023a). We are also interested in combining our LNS solver with neural diving approaches to accelerate or improve the pre-solving quality.

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

# A    PRIMAL BOUND

We plot the primal bounds in Figure 6. The comparison is not as straightforward as for the primal gap since the primal bound does not change much on some of the datasets. But overall, we can still observe that DIFUSCO-LNS achieves the near-optimal primal bound on most datasets and its primal bound converges much faster.

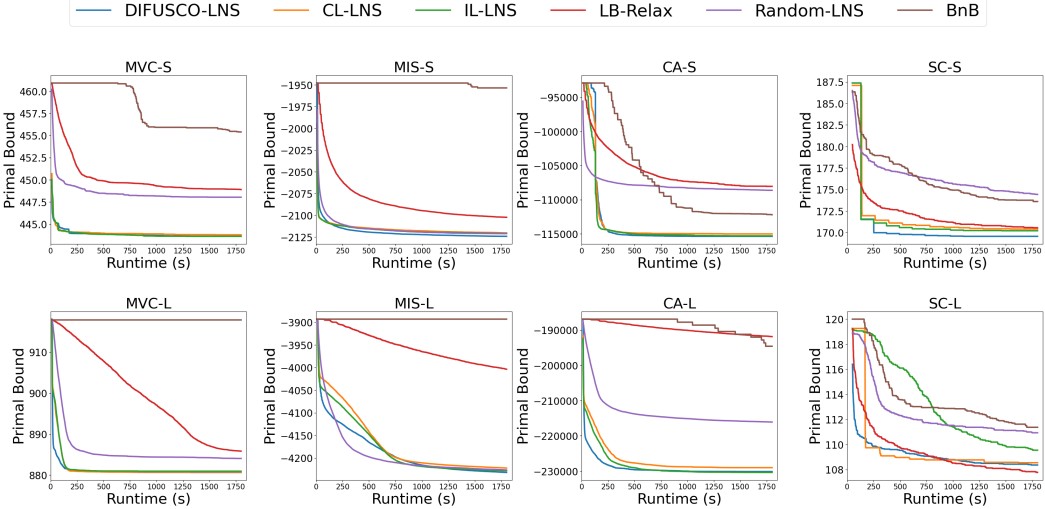

Figure 6: The plot of the primal bound as a function of runtime on all datasets. For a more straight-forward comparison between LNS-based methods and BnB, we clip the initial presolving stage (30 seconds for SC-L and 10 seconds for others) for all methods in the plot. Also, we remove some results from BnB on SC-L whose primal bound found by 1 minute is much larger than the final primal bound. This is only for visualization purposes and not conducted in the calculation of the primal gap and primal integral.

Table 4: Comparative result of all methods in the primal bound (PB) at 30-minute cutoff. The best result is bolded on each dataset.

| Dataset | PB $\downarrow$ MVC-S | PB $\downarrow$ MIS-S | PB $\downarrow$ CA-S | PB $\downarrow$ SC-S |
|---|---|---|---|---|
| BnB | $455.40 \pm 5.64$ | $-1953.42 \pm 8.24$ | $-112205.71 \pm 31.86$ | $173.60 \pm 25.80$ |
| Random-LNS | $448.04 \pm 23.70$ | $-2121.00 \pm 3.01$ | $-108626.00 \pm 13.39$ | $174.42 \pm 25.50$ |
| LB-Relax | $448.91 \pm 23.26$ | $-2102.22 \pm 4.57$ | $-108049.11 \pm 17.60$ | $170.55 \pm 15.26$ |
| IL-LNS | $\mathbf{443.63 \pm 1.50}$ | $-2120.70 \pm 2.55$ | $-115308.56 \pm 9.48$ | $170.20 \pm 12.99$ |
| CL-LNS | $443.81 \pm 2.74$ | $-2120.20 \pm 2.61$ | $-115020.58 \pm 10.69$ | $170.40 \pm 15.91$ |
| DIFUSCO-LNS | $443.65 \pm 1.74$ | $\mathbf{-2124.12 \pm 1.86}$ | $\mathbf{-115375.66 \pm 8.46}$ | $\mathbf{169.55 \pm 14.51}$ |
| Dataset | MVC-L | MIS-L | CA-L | SC-L |
| BnB | $917.87 \pm 6.05$ | $-3893.15 \pm 6.10$ | $-194586.48 \pm 34.81$ | $111.38 \pm 24.62$ |
| Random-LNS | $884.06 \pm 3.68$ | $-4226.42 \pm 2.65$ | $-216093.9 \pm 12.26$ | $110.94 \pm 31.43$ |
| LB-Relax | $885.85 \pm 3.72$ | $-4003.72 \pm 4.44$ | $-191858.19 \pm 16.28$ | $107.77 \pm 17.61$ |
| IL-LNS | $880.95 \pm 2.56$ | $-4228.15 \pm 3.26$ | $\mathbf{-230391.48 \pm 7.19}$ | $109.55 \pm 25.31$ |
| CL-LNS | $880.74 \pm 2.30$ | $-4222.02 \pm 4.07$ | $-229031.93 \pm 10.85$ | $108.55 \pm 23.70$ |
| DIFUSCO-LNS | $\mathbf{880.65 \pm 2.11}$ | $\mathbf{-4231.80 \pm 2.94}$ | $-230142.35 \pm 6.98$ | $\mathbf{108.37 \pm 15.98}$ |

# B    ABALATION ON THE SOURCE OF IMPROVEMENT

Diffusion models are generally more expressive but also suffer from a long generation time. We attribute the success of DIFUSCO-LNS to its inherent compatibility with LNS heuristic. The sup-plementary computational load introduced by its iterative generation is negligible when compared

to the sub-ILP solving time and the time required for preparing the input graph. We first compare the per-iteration running time of CL-LNS and DIFUSCO-LNS in Table 5. It can be seen that except on MIS-S, DIFUSCO-LNS does not show a clear increment in its total running time or the model inference time, where we also consider the time for preparing the input graph a part of the model inference time.

| | MVC-S | | MIS-S | | CA-S | | SC-S | |
|---|---|---|---|---|---|---|---|---|
| | Total | ML | Total | ML | Total | ML | Total | ML |
| CL-LNS | 24.98 | 0.18 | 1.97 | 1.02 | 42.02 | 0.49 | 120.81 | 0.69 |
| DIFUSCO-LNS | 23.01 | 0.18 | 2.83 | 1.90 | 51.29 | 0.50 | 116.36 | 0.71 |

Table 5: Per-iteration LNS running time of CL-LNS and DIFUSCO-LNS on four small datasets. Both the total running time (Total) and the model inference time (ML) for each iteration are presented.

Based on this observation, we further verify that DIFUSCO-LNS learns a better destroy policy than CL-LNS so it brings a larger per-iteration improvement in the primal bound on average. We visualize the comparison in Figure 7. We plot the 10-iteration improvement on MVC-S, CA-S, and SC-S and the 100-iteration improvement on MIS-S since the LNS typically has a much larger number of iterations on MIS-S (greater than 500) than that on other datasets (less than 100). We can also clearly observe that given the same number of iterations, DIFUSCO-LNS achieves a better primal bound, which verifies our assumption that DIFUSCO-LNS learns a better destroy policy.

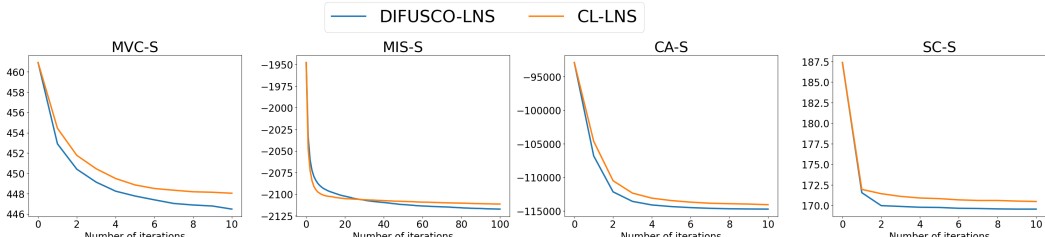

Figure 7: The plot of the primal bound as a function of the number of iterations, averaged over 40 instances. 100 steps are plotted for MIS-S dataset due to its large number of total steps (greater than 500) while 10 steps are plotted for others (total number of steps less than 100).

