# OpenReview forum: "DIFUSCO-LNS: Diffusion-Guided Large Neighbourhood Search for Integer Linear Programming"
_ICLR.cc/2024/Conference — Submitted to ICLR 2024_

### Official Review · Reviewer_Xy75 · 2023-10-30

**Soundness:** 2 fair
**Presentation:** 3 good
**Contribution:** 2 fair
**Rating:** 3
**Confidence:** 4

**Summary:**

The authors propose using a diffusion model for generating neighborhoods for use in large neighborhood search. The diffusion model is trained to imitate local branching, an oracle method that is supposed to find the best local neighborhood to search. The authors present results comparing their approach against other learning-based approaches, heuristics, and branch and bound. They demonstrate mixed results with learning approaches tending to outperform fixed heuristics.

**Strengths:**

The main strengths of the work are that it evaluates against several baselines, using several metrics, and in several ILP settings including generalization to larger instances. Additionally, the paper overall gives a reasonable explanation of the model architecture, data collection, and settings.

**Weaknesses:**

The main weakness of the work is that it seems to just be application of a diffusion model to improve LNS solving without further developing or integrating any of the ideas in diffusion or LNS to get improved performance. For instance, previous approaches generally consider directly predicting the neighborhood. However, given a generative diffusion model one might consider generating several neighborhoods, evaluating them, and selecting the best. Otherwise, you might also consider integrating optimization-based objectives in the diffusion model to guide the diffusion model towards generating better solutions.

Additionally, while the authors do evaluate many metrics, the performance difference between the proposed approach and previous work seems to be quite small and within the uncertainty intervals. It seems unclear that the proposed approach does improve over previous work but in the case that it generally does improve just within a small margin, you might consider evaluating win rate across instances or computing an average rank across instances to understand which algorithm generally solves the problem the fastest.

The problem instances also seem to be effectively solved quite quickly with the primal gap quickly reaching below 10-3. It would help to explain what level of primal gap is reasonable for these instances to be considered solved.

**Questions:**

The authors should consider some rephrasing to better situate their work within the context of optimization.
Positioning of the work:
1st paragraph, last sentence
Our work … belongs to the category of approximate solvers.
It seems that this work doesn’t give approximation guarantees and also doesn’t give indication of how close to optimality a solution is so instead should be considered a primal solver.

3rd paragraph
It is unclear why the referenced LNS approaches require domain-expert knowledge. It seems that they all readily take the ILP formulation as input without much tuning (other than selecting the neighborhood size which is needed in this work as well).
In the same paragraph, it is not obvious why a data-driven approach is necessarily a better alternative as it assumes access to a distribution of problem instances and requires offline training. Here it might be helpful to give a high-level explanation of why learning-based methods should work well.

It would be helpful to explain why local branching is something we desire to learn, does it give performance guarantees for LNS? Does it empirically work well but is just too slow?

Average rank seems to be computed using the summary statistics. However, it might be informative to include a metric measuring the average rank that averages over problem instances. This would help give an idea of whether a given algorithm was generally solving problems faster overall.

Small errors:

2nd paragraph
Heuristics … is called … -> heuristics are called
3rd paragraph
Hand-craft destroy -> hand-crafted destroy
P3 local branching paragraph
“Can we” -> delete this, or missing rest of sentence?

---

> ### Author Response · Authors · 2023-11-23
> **Reply to Reviewer Xy75**
>
> We want to thank Reviewer Xy75 for your valuable suggestions. We will polish the paper accordingly. Here is the response to your questions.
>
> > The main weakness of the work is that it seems to just be application of a diffusion model to improve LNS solving without further developing or integrating any of the ideas in diffusion or LNS to get improved performance.
>
> Please see the first part of our general response.
>
> > The problem instances also seem to be effectively solved quite quickly with the primal gap quickly reaching below 10-3. It would help to explain what level of primal gap is reasonable for these instances to be considered solved.
>
> We compute the primal gap based on the optimal primal bound among all methods in our experiment, therefore, reaching $10^{-3}$ does not really mean the model reaches the optimal value. Also please see our general responses on this.
>
> > It would be helpful to explain why local branching is something we desire to learn, does it give performance guarantees for LNS? Does it empirically work well but is just too slow?
>
> LB (local branching) can give a guaranteed optimal neighborhood for LNS if it can be solved optimally, but it is NP-complete and shares the same time complexity as the original problem. Therefore, it is usually not used when the primal heuristics are needed within a short time.
>
> > It seems that this work doesn’t give approximation guarantees
>
> Thanks for pointing this out. We have changed the phrase from approximate solvers to primal solvers.
>
> > Here it might be helpful to give a high-level explanation of why learning-based methods should work well.
>
> Thanks for the insightful comment. We have added a paragraph explaining that.
>
> > Average rank seems to be computed using the summary statistics. However, it might be informative to include a metric measuring the average rank that averages over problem instances. This would help give an idea of whether a given algorithm was generally solving problems faster overall.
>
> This is a really helpful suggestion. We have made the corresponding change in Table 2 and 3. From the current average rank, CL-LNS looks stronger which is also consistent with the conclusion in the original paper.

---

### Official Review · Reviewer_Gizx · 2023-10-31

**Soundness:** 3 good
**Presentation:** 2 fair
**Contribution:** 3 good
**Rating:** 5
**Confidence:** 3

**Summary:**

The paper proposes a diffusion-based approach for learning a large neighborhood heuristic for solving integer linear programming problems (ILPs). Large neighborhood primal heuristics can be used to quickly find high-quality solutions to ILPs using commercial ILP solvers. They do so by iteratively destroying (using a destroy heuristic) and repairing (using the ILP solver) a given starting solution. In recent years, a number of machine learning approaches have been proposed that aim to learn good destroy heuristics. The proposed method builds upon earlier works (using the same model architecture, features, and data collection) but uses a diffusion-based learning scheme. The authors evaluate their approach on 4 problems and observe that it outperforms earlier approaches.

**Strengths:**

- The idea to use a diffusion model for learning destroy heuristics is novel. Furthermore, the use of a diffusion model is well motivated and straightforward.
- The proposed approach shows good performance and outperforms all other evaluated approaches.

**Weaknesses:**

- In my opinion, the paper is lacking additional ablation studies or experiments that evaluate the influence of the different hyperparameters. The authors only conduct one experiment that analyzes the effect of the number of diffusion steps. They report results for the values 5, 10, 20, 50 and find that 50 steps lead to the best results. This raises the question if the performance of the approach can be further improved by selecting an even higher number of steps. Overall, the relationship between number of diffusion step, prediction time, and prediction quality could be explored further. Fig. 4, which shows the results for the existing experiments, is also a bit difficult to read. Furthermore, the influence of other hyperparameters (e.g., destroy size) could be evaluated further.
- The novelty of the paper is very slightly limited by the fact that the authors use the same model architecture, features, data collection etc. as earlier work. The main contribution of the authors is that they replace the imitation/contrastive learning approach of earlier works with a diffusion-based approach.
- The quality of Fig. 1 and Fig. 2 could be improved. For example, by using the same font for all Figures. Both figures are also not mentioned or explained in the text. For Fig. 2 it is not clear what elements are added and why only x_1 and x_n are considered on the right hand side (and not x_2, x_3, …).
- It is not clear based on which (near-)optimal values the primal gap is calculated. Ideally, the authors would also report the primal bound in the Appendix to make comparisons for future works easier.
- There are some minor spelling mistakes etc. (LSN instead of LNS (page 2), Arechitecture (Fig. 2), unnecessary comma at the end of the baselines paragraph. etc).

**Questions:**

- The approach uses a larger number of hyperparameters (often different parameters for different problems). How have these been selected?

---

> ### Author Response · Authors · 2023-11-23
> **Reply to Reviewer Gizx**
>
> We want to thank Reviewer Gizx for your valuable suggestions. We will polish the paper accordingly. Here is the response to your questions.
>
> > In my opinion, the paper is lacking additional ablation studies or experiments that evaluate the influence of the different hyperparameters.
>
> The LNS-related hyperparameters are set to the same as used in the previous work (CL-LNS, Huang et al., 2023). For diffusion related parameters, please see Figure 5. Our main finding is that different types of problems have different best diffusion steps and diffusion schedules. For example, a linear schedule with 10 steps works best for MIS problems, while a cosine schedule with 2 steps works best for CA problems.
>
> > The novelty of the paper is very slightly limited by the fact that the authors use the same model architecture, features, data collection etc. as earlier work.
>
> Please see the first part in our general response.
>
> > It is not clear based on which (near-)optimal values the primal gap is calculated. Ideally, the authors would also report the primal bound in the Appendix to make comparisons for future works easier.
>
> Please see the second part in our general response. The optimal values are chosen as the best primal bound among all methods, including all baselines and DIFUSCO-LNS. The primal bound is reported in Figure 5 in Appendix.
>
> > The quality of Fig. 1 and Fig. 2 could be improved.
>
> Thanks for pointing this out! We have improved the quality of the figures.

---

### Official Review · Reviewer_3tnY · 2023-11-01

**Soundness:** 3 good
**Presentation:** 3 good
**Contribution:** 2 fair
**Rating:** 5
**Confidence:** 4

**Summary:**

The paper presents an ML-guided LNS framework for MIPs. It uses a diffusion model to guide variable selection in the destroy step of LNS. The variable selection is treated as a generative task, and is learned by imitating the Local Branching expert policy. In the experiment, the new method DIFUSCO-LNS is compared against a variety of ML-guided approaches and heuristic approaches. The presented results show that the proposed method finds better solutions at a faster speed on most benchmarks.

**Strengths:**

1. Applying diffusion is a novel idea and an interesting direction for LNS mip solving.

2. Experimental results show promise for the approach.

**Weaknesses:**

1. The results are promising on some benchmarks but overall not super impressive. Also, could you highlight the innovation in diffusion models from this paper that enables its application for LNS?

2. It would be interesting to see the comparison prediction accuracy / per-iteration improvement to confirm that difusco-LNS is indeed making better predictions.

3. Related to the previous point, It would be good to report the ML inference time overhead during testing. My understanding is that diffusion requires a more expensive denoising process than the other ML approaches using the same architecture.

4. It seems DIFUSCO-LNS is sensitive to hyperparameters. It is not discussed how the best hyperparameters were chosen in the paper.

**Questions:**

1. You mentioned that you were not able to reproduce results for some baselines due to differences in hardware/compute resources. Can you elaborate more on this? For LB-RELAX there seems to be quite a huge difference. From my own experience, sometimes it is due to different software versions (like SCIP or Gurobi). The other time it might be due to hardware differences: a slower machine computes different heuristics at the BnB nodes and thus produces different results if the wall-clock time budget is fixed.

2. Difusco-LNS also takes advantage of multiple good solutions following previous work. I wonder if a contrastive learning component can be built into the diffusion model so that you can leverage bad solutions from LB?

3. I realize the green curve in Figure 4 has an increasing trend at around 500-600 seconds. What happens there?

4. From the ablation studies, it seems that DIFUSCO-LNS is sensitive to a couple of hyperparameters. I wish to understand whether you need to fine-tune them for different benchmarks.

---

> ### Author Response · Authors · 2023-11-23
> **Reply to Reviewer 3tnY**
>
> We want to thank Reviewer 3tnY for your valuable suggestions. We will polish the paper accordingly. Here is the response to your questions.
>
> > Could you highlight the innovation in diffusion models from this paper that enables its application for LNS?
>
> Please see our answer in the general response.
>
> > It would be interesting to see the comparison prediction accuracy / per-iteration improvement to confirm that difusco-LNS is indeed making better predictions.
>
> We verify that DIFUSCO-LNS learns a better destroy policy than CL-LNS so it brings a larger per-iteration improvement in the primal bound on average. We visualize the comparison in Figure 7. We plot the 10-iteration improvement on MVC-S, CA-S, and SC-S and the 100-iteration improvement on MIS-S since the LNS typically has a much larger number of iterations on MIS-S (greater than 500) than that on other datasets (less than 100). We can also clearly observe that given the same number of iterations, DIFUSCO-LNS achieves a better primal bound, which verifies our assumption that DIFUSCO-LNS learns a better destroy policy.
>
> > Related to the previous point, It would be good to report the ML inference time overhead during testing. My understanding is that diffusion requires a more expensive denoising process than the other ML approaches using the same architecture.
>
> Please see our answer in the general response.
>
> > It seems DIFUSCO-LNS is sensitive to hyperparameters. It is not discussed how the best hyperparameters were chosen in the paper.
> > From the ablation studies, it seems that DIFUSCO-LNS is sensitive to a couple of hyperparameters. I wish to understand whether you need to fine-tune them for different benchmarks.
>
> The LNS-related hyperparameters are set to the same as used in the previous work (CL-LNS, Huang et al., 2023). For diffusion related parameters, please see Figure 5.
>
> > You mentioned that you were not able to reproduce results for some baselines due to differences in hardware/compute resources. Can you elaborate more on this?
>
> In our research, we diligently followed the methodologies outlined in the publicly available code from the previous study (found at https://github.com/facebookresearch/CL-LNS). However, during our replication efforts on our own hardware, we observed variations in the results. Specifically, the trend of the performance curve differed from the original findings. This discrepancy could be attributed to differences in hardware and computing resources between our setup and that of the original study.
>
> > I wonder if a contrastive learning component can be built into the diffusion model so that you can leverage bad solutions from LB?
>
> Good question.  Although we currently have focused on improving the  generative probability of good solutions with diffusion, we do believe that contrastive learning [1], especially contrastive generative modeling [2], is an important future direction.  Thanks for this suggestion.
>
> > I realize the green curve in Figure 4 has an increasing trend at around 500-600 seconds. What happens there?
>
> Good catch! We found this is due to the solver on an instance accidentally stopped earlier. We have fixed the curve.
>
> [1] Huang, Taoan, et al. "Searching large neighborhoods for integer linear programs with contrastive learning." International Conference on Machine Learning. PMLR, 2023.
>
> [2] Rafailov, Rafael, et al. "Direct preference optimization: Your language model is secretly a reward model." arXiv preprint arXiv:2305.18290 (2023).

---

### Author Response · Authors · 2023-11-23
**General response**

We are grateful to all reviewers for their insightful comments. We appreciate that reviewers found our method to be novel (3tnY, Gizx) / well motivated (Gizx), and the results to be promising (3tnY, Gizx).

Several new experiments and analyses, as per your suggestions, have been incorporated, primarily in the appendices. Additionally, we have addressed the  questions of each individual reviewer with the shared clarification points below:

### Novelty of Diffusion Model-based LNS.

We want to highlight that diffusion models are naturally good choices for modeling LNS heuristics but have not been investigated before for this problem. Our work shows that diffusion models are more expressive (at expressing multimodal distributions) than previous single-shot prediction models used in IL-LNS, RL-LNS, and CL-LNS. Although they can be slow during the iterative generation, this drawback is almost negligible in LNS since the time for solving subproblems in MILP is much longer than the computation time of the diffusion part, as shown in our Table 5.  These fundings, we believe, significantly enhance the understanding of the power of diffusion models in effective learning of  LNS heuristics, and for better solving MILP, one of the most fundamental challenges in computer science and machine learning.

### Clarification on the Primal Gap.

We used SCIP to solve each problem instance for 1 hour, but it did not give a better bound than the ones achieved by the neural method in 30 minutes. Therefore, we simply used the best primal bound achieved by all methods, including the baselines and DIFUSCO-LNS, as the optimal primal bound to compute the primal gap. In this case, the primal gap will become 0 for the best neural method on each instance, but this does not mean that the neural method always achieves the optimal performance. We have made this point more clear in our revision, thanks for the reviewer’s question.  We also report the primal bounds in Figure 5 of Appendix.

---

### Meta-Review · Area_Chair_PqB8 · 2023-12-08

**Metareview:**

The authors use diffusion models to guide neighborhood search for integer linear programming problems, and conduct experiments on the open source SCIP solver demonstrating modest benefits relative to prior work on several benchmark datasets. While the idea of using a diffusion model for this task is interesting, the resulting improvements do not provide sufficient evidence for the claims made in the paper, and the work also does not provide much analysis on why diffusion models might do well at this task. Hence, I recommend rejection.

**Justification For Why Not Higher Score:**

Empirical paper without sufficiently strong empirical results.

**Justification For Why Not Lower Score:**

N/A

---

### Decision · Program_Chairs · 2024-01-16

Reject